# Experiences of and response to the COVID-19 pandemic at private retail pharmacies in Kenya: a mixed-methods study

Peter Mwangi Mugo [1,2] Audrey Mumbi,[1] Daniella Munene,[3] Jacinta Nzinga [1,2] Sassy Molyneux,[2,4] Edwine Barasa,[1,4] PSK COVID-19 Response Taskforce[5]

¹Health Economics Research Unit, KEMRI-Wellcome Trust Research Programme, Nairobi, Kenya
²Health Systems and Research Ethics Department, KEMRI-Wellcome Trust Research Programme, Kilifi, Kenya
³CEO Office, Pharmaceutical Society, Kilifi, Kenya
⁴Nuffield Department of Medicine, University of Oxford, Oxford, UK
⁵COVID-19 Response Taskforce, Pharmaceutical Society of Kenya (PSK), Nairobi, Kenya

**Correspondence to**
Dr Peter Mwangi Mugo;
PMugo@kemri-wellcome.org

## ABSTRACT

**Objectives** To assess experiences of and response to the COVID-19 pandemic at community pharmacies in Kenya.

**Design, setting and participants** This was a mixed-methods study conducted from November 2020 to April 2021, targeting service providers in three counties (Nairobi, Mombasa and Kisumu), selected purposively to represent the main urban centres; pharmacies were selected randomly from a list of licensed pharmacies.

**Results** Of 195 sampled pharmacies, 108 (55%) completed a questionnaire and 103 (53%) received a simulated client call; 18 service providers were interviewed. The initial weeks of the pandemic were characterised by fear and panic among service providers and a surge in client flow. Subsequently, 65 (60%) of 108 pharmacies experienced a dip in demand to below prepandemic levels and 34 (31%) reported challenges with unavailability, high price and poor quality of products. Almost all pharmacies were actively providing preventive materials and therapies; educating clients on prevention measures; counselling anxious clients; and handling and referring suspect cases. Fifty-nine pharmacies (55% (95% CI 45% to 65%)) reported receiving a client asking for COVID-19 testing and a similar proportion stated they would support pharmacy-based testing if implemented. For treatment of simulated clients, most pharmacies (71%, 73 of 103) recommended alternative therapies and nutritional supplements such as vitamin C; the rest recommended conventional therapies such as antibiotics. While 52 (48%) of 108 pharmacies had at least one staff member trained on COVID-19, a general feeling of disconnection from the national programme prevailed.

**Conclusions** Private pharmacies in Kenya were actively contributing to the COVID-19 response, but more deliberate engagement, support and linkages are required. Notably, there is an urgent need to develop guidelines for pharmacy-based COVID-19 testing, a service that is clearly needed and which could greatly increase test coverage. Pharmacy-based COVID-19 programmes should be accompanied with implementation research to inform current and future pandemic responses.

## STRENGTHS AND LIMITATIONS OF THIS STUDY

⇒ The study used multiple quantitative and qualitative methods to enhance the comprehensiveness and reliability of the findings.
⇒ We used simulated clients to objectively assess provider practices, an approach that is gaining acceptance as a gold standard for the measurement of clinical practice quality around the world.
⇒ The study took advantage of information collected in a recent study to speed up recruitment and data collection.
⇒ One weakness of the study was the low response rate (55%), considering that it targeted pharmacies that had participated in a similar study less than a year previously.

## INTRODUCTION

The COVID-19 pandemic is an ongoing global health emergency.[1 2] As of September 2021, over 230 million cases had been confirmed globally resulting in over 4.7 million deaths.[3] For the better part of 2020, non-pharmaceutical interventions such as early case detection, contact reduction and hygiene measures were the main prevention strategies.[4 5] In early 2021, following unprecedented public–private collaboration and speed in research and development, a number of highly effective vaccines became available.[6 7]

Non-pharmaceutical supportive care remains the key approach to treatment and management.[8–10] Among hospitalised patients, pharmaceuticals that have been shown to be effective include corticosteroids such as dexamethasone, antivirals such as remdesivir and immunomodulators such as tocilizumab.[8 11 12] In the outpatient setting, pharmaceuticals used to manage mild COVID-19 cases include those targeting fever and pain such as ibuprofen, and those aiming

to relieve respiratory congestion.[9 12] Other more specific agents that have been tried, such as hydroxychloroquine, ivermectin and azithromycin, are discouraged.[9 12] The WHO and the US National Institutes of Health recommend that such unproven therapies be used only within the context of clinical trials.[13 14]

It is predicted that follow-up waves of COVID-19 infection may continue indefinitely.[15] Strategies are, therefore, required to enhance and sustain coverage, cost-effectiveness, and efficiency of prevention and treatment interventions. Private retail pharmacies (also referred to as community pharmacies) are a unique channel through which COVID-19 prevention and treatment interventions can be delivered, since they are often the first or only point of contact with the healthcare system in developing countries.[16–18] The main reasons for preferential care-seeking at pharmacies, compared with health facilities, include greater accessibility, lower cost and greater perceived privacy.[19 20] The International Pharmaceutical Federation highlights key areas where the pharmacy sector can support the pandemic response, including: management of the medication supply chain, patient education, identification and referral of patients to other healthcare providers, and provision of prevention materials and therapies.[21 22] During the COVID-19 pandemic, pharmacies in many settings, including Kenya, were categorised as essential services hence they were exempted from closure.[23]

The first COVID-19 case in Kenya was reported on 13 March 2020, and as of 12 October 2021, there were 251 248 confirmed cases and 5190 deaths.[24] A number of control measures have been instituted by the government at different times depending on rates of infection and mortality.[23 25 26] Vaccine roll-out in Kenya started in March 2021.[27] The aim of the study was to assess the preparedness, experiences and response to the COVID-19 pandemic at private retail pharmacies in Kenya, with an overall goal of identifying strategies for maximising the contribution of pharmacies to the national response.

## METHODS
### Study design
We conducted a cross-sectional descriptive study, including a questionnaire survey, a simulated client (SC) survey and in-depth interviews. A mixed-methods approach was used in data collection and analysis: insights from interviews were used to refine the survey tools and probe quantitative data; and survey data were used to refine the interview guide and qualitative data analysis. Given the physical distancing requirements at the time, data collection was done remotely.

### Study settings and populations
Community pharmacies in Kenya are generally small to medium-sized businesses providing typical pharmacy services such as filling of prescriptions, over-the counter products and point-of-care (POC) testing or self-testing

kits for common diseases. While traditionally a fragmented market, the sector is starting to consolidate. In 2019, of 5033 licensed pharmacies, 4286 (85%) were independent (single branch) pharmacies, 594 (12%) belonged to small chain networks (2–5 branches) and 152 (3%) were part of large chain networks (6–41 branches) belonging to 13 companies.[28]

The study was conducted in three counties (Nairobi, Mombasa and Kisumu), selected purposively to represent the main urban centres, where there was high concentration of pharmacies and COVID-19 cases. As of June 2020, the three counties together contributed 32% (1602/5033) of licensed pharmacies[28] and 78% (2229/2,862) of reported COVID-19 cases.[24 29]

We targeted pharmacies that had participated in a pharmacy HIV prevention (PHP) study about a year previously. Pharmacies had been selected through stratified random sampling where the required number of pharmacies for each county was picked by an independent data manager from the list of licensed pharmacies for that county using the Excel Rand() function. The sample size for the PHP study (n=195) had been calculated to provide 80% power to detect associations at the 5% significance level, assuming a binary dependent variable split at 70% and 50% across a binary predictor variable. The sample size achieved in the current study (n=108) gives 57% power for similar parameters and assumptions, and 80% power for a 75%–50% split. Respondents were pharmacists (bachelor's degree holders) and pharmaceutical technologists (diploma holders). The preferred questionnaire respondent was the pharmacy-in-charge, the most experienced staff member or the most highly educated, in that order. Interview participants were purposively selected from among questionnaire respondents, targeting those that showed enthusiasm for the study through prompt and detailed responses. We aimed to increase variability of interview data by stratifying the sample by county, practice setting (urban vs rural) and gender (male vs female).

### Data collection
Data collection started in November 2020 (approximately 7 months after the pandemic reached Kenya) and was concluded in December 2020 (for the surveys) and April 2021 (for the interviews).

We used previously obtained phone numbers to contact sampled pharmacies, obtain informed consent and share a link to an online questionnaire. One questionnaire, taking about 20–30 min, was completed by each participating pharmacy (online supplemental file A). Up to three reminder phone calls were made. Data on pharmacy characteristics were imported from the PHP study database.

SCs made unscheduled phone calls to participating pharmacies, mimicking a client seeking COVID-19 preventive therapies (online supplemental file B). The SCs (two women and two men, aged 25–35 years) had previously been engaged in the PHP study and trained comprehensively over a 2-week period. For the current

study, further training was provided over a 1-week period through the Teams digital meeting platform. An online debrief questionnaire was used to record observations (online supplemental file C). All calls were made on weekdays (Monday–Friday) between 8:00 and 17:00 hours, over a 3-week period. Median (range) duration of calls was $4^{1-12}$ min. The majority (68%) of calls were made to a publicly available phone number, while the rest used a number in the study database as there was no public number. A few weeks after the simulated call, participating pharmacies were contacted to check if they suspected any clients visiting the pharmacies during the survey period to be 'fake' (detection survey).

In-depth interviews were conducted by the first and second authors using a semistructured interview guide (online supplemental file D). Interviews were conducted via the Teams digital meeting platform as a first option or a simple phone call if the participant so preferred or if internet connectivity was poor.

## Data management and analysis

Quantitative data were captured directly into an open-source database (REDcap, Vanderbilt University). Data cleaning and analysis was carried out using Stata V.15. Summary statistics were compared across counties using the $\chi^2$ test or analysis of variance for means. Exploratory multivariable modelling was done using logistic regression. Variables with p<0.10 in bivariate analysis were included in the multivariable model and those with p<0.05 in the multivariable model were considered to be statistically significant. Systematicity in detection was assessed by correlating detection status with caller, pharmacy characteristics and simulation process features.

Qualitative interviews were audiorecorded and transcribed verbatim in Microsoft Word, translating into English where necessary. To ensure quality, all transcripts were cross-checked against the audiorecordings. Transcripts were imported into NVivo V.12 (SR International, Australia) and coded by the first and second authors. Analysis followed a thematic approach with initial categories based on the interview guide and emergent themes integrated in a second round of coding. A checklist on Consolidated criteria for Reporting Qualitative research was completed (online supplemental file E).

## Patient and public involvement

An early version of the study protocol received inputs from the Pharmaceutical Society of Kenya (PSK) COVID-19 response taskforce composed of pharmacy sector stakeholders. Preliminary findings were shared with the pharmacy sector at the 2021 PSK Annual Conference and a policy brief has been distributed to stakeholders in the ministry of health, pharmacy sector and other implementing partners. Findings were also discussed in a local Radio show in February 2022.

## RESULTS

### Response rate and characteristics of participants

Of 195 target pharmacies, 108 (55%) completed a questionnaire. Participating pharmacies were generally similar to non-participating pharmacies, but had features indicative of higher operational capacity and interest in public health interventions, such as having a computerised stock management system, written job descriptions and providing blood pressure measurement (data not shown). Ninety-four (87%) pharmacies were located in urban areas; 44 (41%) had a consultation room; 71 (67%) had a computerised stock management system and 13 (12%) were part of a chain network (online supplemental table Ia). The majority of questionnaire respondents were male, below 40 years, and had a pharmacy diploma (online supplemental table Ib).

One hundred and three (95%) pharmacies received a SC call. Thirteen (13%) calls were determined as probably detected. Detection was less likely if the call was made by one particular caller (perfect prediction), gender of person answering the call was male (adjusted OR, aOR 0.1 (95% CI 0.0004 to 0.4), p=0.01); the pharmacy had a publicly available phone number (aOR 0.01 (95% CI 0.0006 to 0.4), p=0.01), was part of a chain network (perfect prediction), opened 24 hours daily (perfect prediction) or had an on-site laboratory (perfect prediction). Main call outcomes (preventive therapies recommended, options for delivery of recommended products and referral options) did not vary by detection status (data not shown), therefore all data were included in analysis.

Of 18 interview participants, 11 (61%) were male, 16 (88%) were below 40 years, 11 (61%) were pharmacy owner or in-charge, and 12 (67%) were diploma holders. Number of participants was balanced by county (6 each), but was skewed towards urban setting (13 vs 5) and male gender (11 vs 7) (online supplemental table II; the numbers in the participant list can be used to cross-reference to the numbers included with illustrative quotes below).

### Impact of the pandemic on service providers and pharmacy operations

Interview participants described fear and panic when they first heard about COVID-19, especially regarding the risk of exposure to infection:

> In the beginning, it sent everybody into a panic mode. Including my entire staff where everybody now thought this is a killer disease and being healthcare workers, they must be people who will be among the victims.
>
> - Male pharmaceutical technologist, Nairobi, 105

Some providers were in denial, feeling that the pandemic would never reach 'where we are', and others reported that some clients believed the pandemic was

*'just an avenue maybe for our politicians ready to make some money or something'*.

The main concerns reported in the questionnaire were: exposure to infection at work (66% of respondents), 'keeping my family safe' (22%), meeting financial needs (6%), and being caught up in the curfew (3%).

One hundred (93%) of 108 pharmacies reported pandemic-induced operational challenges (figure 1). The main supply-side challenge was unavailability and high price of products and the main demand-side challenge was reduced client flow.

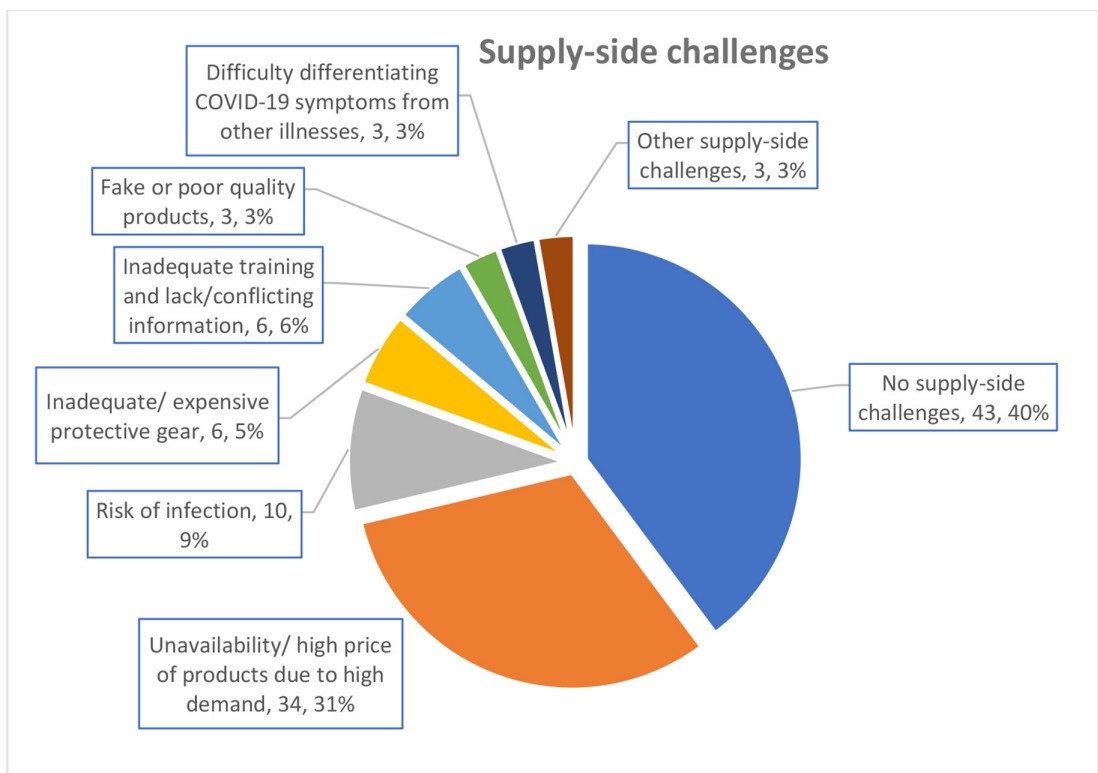

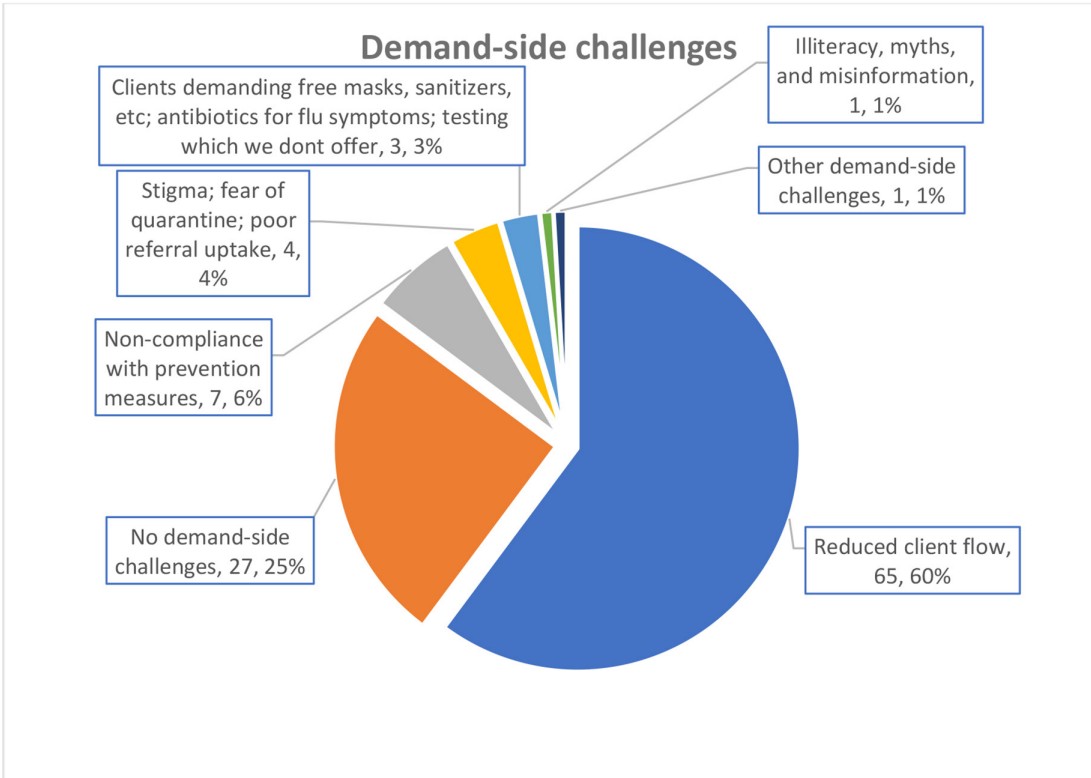

**Figure 1** Challenges experienced by pharmacies during the COVID-19 pandemic in Kenya.

Interview data corroborated survey findings. Interviewees described temporal changes in client flow, with an initial surge of clients buying vitamins and other immune boosters, followed a few weeks later by a dip in demand to below pre-pandemic levels. The initial surge was accompanied by an increase in self-medication and stockpiling of long-term medications, such as for diabetes. While some of these requests were deemed potentially inappropriate, providers were not always able to effectively intervene:

> They just keep [these drugs] in case. …Like there was one [client] who came with a running nose, and then he was telling me he wanted Zithromax [a brand of the antibiotic azithromycin]. 'Get Flu gone, get antihistamine, this one will help you' [I told him]. So sometimes as a primary healthcare provider, you try to advise them, but now they have made up their mind, so you just go ahead and let them go [somewhere else]. Because whether you refuse, you do what, they've made up their mind.
>
> - Male pharmaceutical technologist, Kisumu, 303

Participants attributed the subsequent reduction in demand mainly to the dusk-to-dawn curfew, despite the fact that pharmacy providers were classified as essential workers and issued curfew passes:

> Of course, the curfew really beat us down, because now by the time [it's] seven o'clock, people are going home… and it's our prime time to serve clients as from 7 pm. So, we as well missed out on business.
>
> - Male pharmacy owner, Mombasa, 203

A number of interviewees believed that the decrease in service utilisation was due to fear of infection and quarantine among clients:

> Then people shied off from coming to the pharmacy especially the ones with coughs and flus because they were just thinking that I could tell them to go to hospital for the corona tests. So most clients decided to stay at home, not to come to the pharmacy for consultation, or they just sent somebody to buy them the drugs
>
> - Female pharmaceutical technologist, Kisumu, 304

One interviewee saw the stockpiling by clients in positive light, observing that it enabled him to offload slow-moving products:

> People resorted to scavenging where they might find these products and so that way [one was] able to liquidate what was lying in your shops
>
> - Male pharmaceutical technologist, Nairobi, 105

A reduction in hygiene-related illnesses was also hailed as a positive public health effect of the COVID-19 preventive measures:

> Due to proper maintenance of hygiene, the short-short illnesses have reduced. Washing of hands and stuff.
>
> - Male pharmacist, Nairobi, 103

### Adjustments to pharmacy operations

Pharmacies adjusted their operations to cope with these challenges and to accommodate the government-instituted prevention measures: 74% reduced opening hours (mean 3 hours (range 1–7)); 38% reduced staffing; and 30% changed their meeting routine (table 1). Adjustments did not vary significantly across counties.

### COVID-19-related products and services provided by pharmacies

Table 2 presents the COVID-19-related products and services that were being provided or requested at pharmacies. Equipment and materials in highest demand were medical masks, N95/K95 respirators, face shields, hand sanitiser and thermometers; while the medications in highest demand were vitamin C, multivitamins and antibiotics. Only 24 (22%) pharmacies reported keeping records related to COVID-19 services.

Interview data suggested that many of these products and services were newly introduced following the outbreak, including products like vitamin C and surgical masks that one would have expected to find routinely in a pharmacy.

### Demand for COVID-19 testing and provider views towards pharmacy-based testing

Fifty-nine (55% (95% CI 45% to 65%)) of 108 pharmacies reported ever receiving a client asking for COVID-19 testing, with a median (IQR) number of requests in the last week of 2.[1–4] At the time of the survey, none of the pharmacies was providing COVID-19 testing; and among the 59 pharmacies experiencing demand for testing, alternatives given to clients included referral to other testing sites (94%) and body temperature measurement (15%). Referral options included: public health facilities (92%), private health facilities (27%) and private labs (13%).

Interviews revealed that test requests were mostly by clients with symptoms, those who suspected they had been exposed, and those who needed certification for travel or employment purposes (eg, drivers and hotel staff).

When asked if they thought COVID-19 testing should be provided in pharmacies, 60% (95% CI 50% to 70%) said 'yes'. Among the 64 that were supportive of pharmacy-based testing, main reasons for support were: pharmacies are easily accessible (36%), clients perceive pharmacies as more confidential (23%), clients shy away from health facilities (19%), and demand exists (9%).

In interviews, the confidentiality and convenience of the pharmacy setting was reiterated, with some interviewees suggesting home-based self-testing as an option:

> If it [testing] is available in pharmacies, somebody can just take the kit, you can just go home, you test

**Table 1** Adjustments to pharmacy operations following the outbreak of COVID-19 in Kenya

| Change | Nairobi (N=38) | | Mombasa (N=36) | | Kisumu (N=34) | | Total (N=108) | |
|---|---|---|---|---|---|---|---|---|
| | n | % | n | % | n | % | n | % |
| **Opening hours*** | | | | | | | | |
| No change | 7 | **18** | 8 | **22** | 10 | **29** | 25 | **23** |
| Reduced open hours per day | 31 | **82** | 25 | **69** | 24 | **71** | 80 | **74** |
| Reduced open days per week | 1 | **3** | 4 | **11** | 0 | **0** | 5 | **5** |
| Closed temporarily | 0 | **0** | 0 | **0** | 1 | **3** | 1 | **1** |
| **Staffing** | | | | | | | | |
| No change | 19 | **50** | 25 | **71** | 19 | **59** | 63 | **60** |
| Reduced no of staff | 18 | **47** | 9 | **26** | 13 | **41** | 40 | **38** |
| Increased no of staff | 0 | **0** | 1 | **3** | 0 | **0** | 1 | **1** |
| Reduced pay | 1 | **3** | 0 | **0** | 0 | **0** | 1 | **1** |
| **Stock volume levels** | | | | | | | | |
| No change | 12 | **32** | 13 | **36** | 8 | **24** | 33 | **31** |
| Reduced overall | 8 | **21** | 12 | **33** | 12 | **35** | 32 | **30** |
| Increased overall | 14 | **37** | 10 | **28** | 14 | **41** | 38 | **35** |
| Both reduced and increased for different items | 4 | **11** | 1 | **3** | 0 | **0** | 5 | **5** |
| **Meeting routine** | | | | | | | | |
| No change | 24 | **63** | 30 | **83** | 22 | **65** | 76 | **70** |
| Resorted to virtual/remote/online meetings† | 8 | **21** | 3 | **8** | 7 | **21** | 18 | **17** |
| Reduced frequency of meetings | 4 | **11** | 1 | **3** | 3 | **9** | 8 | **7** |
| Stopped meetings altogether | 2 | **5** | 1 | **3** | 0 | **0** | 3 | **3** |
| Reduced duration/no of people | 0 | **0** | 1 | **3** | 1 | **3** | 2 | **2** |
| Changed timing | 0 | **0** | 0 | **0** | 1 | **3** | 1 | **1** |

*Three pharmacies in Mombasa and one in Nairobi reduced their open days per week by 1, while one pharmacy in Mombasa reduced by 3 days. One pharmacy in Kisumu closed temporarily for 2 weeks.
†Main virtual platforms used included Zoom, Google Meet and Whatsapp.

yourself. If you find you are positive, you can just self-quarantine, without raising an alarm. So that when the quarantine period is over, you don't get that kind of stigma, rejection.

- Female pharmacy in charge, Mombasa, 201

Among the 42 that were not supportive of pharmacy-based testing, main reasons for opposing were: might be misused (29%), no private space in the pharmacy (24%), risk of infection (15%), protective gear is expensive or not available (12%), there is no demand for the service (10%), and staff are not adequately trained (10%). From the interviews, hesitancy to support pharmacy-based testing was mainly related to lack of capacity ('*I still need a thorough training…*') and fear of infection.

### Handling of SCs seeking COVID-19-related services
When a SC called a pharmacy to ask if they have medicines that the client can use to protect themselves from getting COVID-19, 76 (74%) of 103 pharmacies recommended one or more medications: 27% recommended at least one conventional therapy and 71% at least one

alternative therapy or nutritional supplement (table 3). The main recommendations given to a client suspecting COVID-19 infection were to: get tested, visit the nearest health facility and self-quarantine at home.

### Infection prevention during service provision
The main infection prevention measures during service provision were hand hygiene, face masks, disinfection of services and physical distancing (table 4). Eighty-five per cent of respondents rated the prevention measures instituted at their pharmacy as either adequate (72%) or very adequate (13%). Only one pharmacy reported that a staff member had ever tested positive for COVID-19, though no pharmacy reported regular testing of staff.

### Information sources and training
Sources of general COVID-19 information included: continuing professional development sessions (37% of respondents), ministry of health (MoH) communications (23%), social media and internet (19%), television and radio (12%), Kenya Pharmaceutical Association (KPA)/ PSK communications (5%), scientific journals (2%) and

**Table 2** COVID-19-related products and services provided by private retail pharmacies in Kenya

| Product/service | Nairobi (N=38) N or median | % or IQR | Mombasa (N=36) N or median | % or IQR | Kisumu (N=34) N or median | % or IQR | Total (N=108) N or median | % or IQR |
|---|---|---|---|---|---|---|---|---|
| **Equipment and materials** | | | | | | | | |
| Cloth masks | 1 | 3% | 5 | 14% | 6 | 18% | 12 | 11% |
| No of clients in past week | 20 | 20–20 | 20 | 4–30 | 10 | 5–15 | 15 | 5–25 |
| Medical masks | 37 | 97% | 33 | 92% | 30 | 88% | 100 | 93% |
| No of clients in past week | 50 | 20–100 | 23 | 10–80 | 26 | 20–50 | 30 | 20–70 |
| N95/K95 masks | 24 | 63%* | 18 | 50% | 11 | 32% | 53 | 49% |
| No of clients in past week | 5 | 4–16 | 5 | 2–30 | 1 | 0–2 | 5 | 2–10 |
| Goggles | 6 | 16% | 2 | 6% | 3 | 9% | 11 | 10% |
| No of clients in past week | 0 | 0–10 | 6 | 2–10 | 1 | 0–5 | 1 | 0–10 |
| Face shields | 19 | 50%* | 10 | 28% | 8 | 24% | 37 | 34% |
| No of clients in past week | 1 | 0–6 | 1 | 0–3 | 1 | 0–8 | 1 | 0–5 |
| Gowns/coveralls | 1 | 3% | 1 | 3% | 3 | 9% | 5 | 5% |
| No of clients in past week | 0 | 0–0 | 10* | 0–10 | 0 | 0–1 | 0 | 0–1 |
| Hand sanitiser | 37 | 97% | 35 | 97% | 31 | 91% | 103 | 95% |
| No of clients in past week | 10 | 3–30 | 5 | 2–13 | 5 | 2–13 | 5 | 3–15 |
| Thermometers | 27 | 71% | 23 | 64% | 20 | 59% | 70 | 65% |
| No of clients in past week | 2 | 1–5 | 2 | 1–5 | 1 | 0–2 | 2 | 2–3 |
| Pulse oximeters | 10 | 26% | 11 | 31% | 3 | 9% | 24 | 22% |
| No of clients in past week | 1 | 1–3 | 2 | 0–10 | 1 | 0–2 | 2 | 1–3 |
| **Medicines** | | | | | | | | |
| Vitamin C† | 35 | 92%* | 29 | 81% | 23 | 68% | 87 | 81% |
| No of clients in past week | 10 | 10–30 | 15 | 10–30 | 5 | 3–11 | 10 | 4–20 |
| Multivitamins† | 29 | 76% | 30 | 83% | 24 | 71% | 83 | 77% |
| No of clients in past week | 10 | 5–19 | 10 | 6–20 | 10 | 7–18 | 10 | 6–20 |
| Traditional/ alternative/ natural remedies†, ‡ | 1 | 3% | 2 | 6% | 0 | 0% | 3 | 3% |
| No of clients in past week | 0 | 0–0 | 13 | 5–20 | 0 | 0–0 | 5 | 0–20 |
| Hydroxychloroquine/ chloroquine | 8 | 21% | 7 | 19% | | 6% | 17 | 16% |
| No of clients in past week | 3 | 2–7 | 1 | 0–5 | 3 | 0–5 | 2 | 0–5 |
| Antibiotics†, § | 26 | 68% | 22 | 61% | 23 | 68% | 71 | 66% |
| No of clients in past week | 3 | 2–10 | 15* | 10–39 | 15 | 5–29 | 10 | 3–20 |
| Steroids† | 16 | 42% | 16 | 44% | 16 | 47% | 48 | 44% |
| No of clients in past week | 3 | 1–8 | 9 | 5–19 | 10 | 4–15 | 7 | 3–15 |
| Other medicines¶ | 3 | 8% | 4 | 11% | 2 | 6% | 9 | 8% |
| No of clients in past week | 3 | 1–5 | 28 | 23–45 | 3 | 1–4 | 13 | 3–28 |
| **Information and counselling** | | | | | | | | |
| How to protect themselves from infection | 28 | 74% | 28 | 78% | 27 | 79% | 83 | 77% |
| No of clients in past week | 5 | 2–13 | 5 | 3–15 | 10 | 3–20 | 5 | 3–15 |
| Symptoms to look out for if they suspect infection | 32 | 84% | 28 | 78% | 26 | 76% | 86 | 80% |
| No of clients in past week | 3 | 2–10 | 5 | 4–10 | 9 | 2–11 | 5 | 2–11 |
| What to do if they develop symptoms/ suspect infection | 22 | 58% | 21 | 58% | 16 | 47% | 59 | 55% |
| No of clients in past week | 3 | 2–7 | 3 | 2–5 | 5 | 3–22 | 4 | 2–10 |

**Table 2** Continued

| Product/service | Nairobi (N=38) | | Mombasa (N=36) | | Kisumu (N=34) | | Total (N=108) | |
|---|---|---|---|---|---|---|---|---|
| | N or median | % or IQR | N or median | % or IQR | N or median | % or IQR | N or median | % or IQR |
| Mask use and related problems | 19 | 50% | 19 | 53% | 20 | 59% | 58 | 54% |
| No of clients in past week | 5 | 2–10 | 5 | 2–35 | 15 | 4–20 | 7 | 2–20 |
| Counselling for anxiety related to getting infected/ becoming sick with COVID-19 | 14 | 37% | 16 | 44% | 13 | 38% | 43 | 40% |
| No of clients in past week | 2 | 1–10 | 3 | 1–10 | 3 | 1–5 | 3 | 1–8 |
| Counselling for anxiety about job/ business loss | 5 | 13% | 7 | 19% | 5 | 15% | 17 | 16% |
| No of clients in past week | 2 | 2–8 | 2 | 1–5 | 5 | 3–8 | 3 | 2–8 |
| **Modes of communication with clients** | | | | | | | | |
| Face to face at the pharmacy | 33 | 87% | 33 | 92% | 34 | 100% | 100 | 93% |
| Phone calls (including WhatsApp calls, etc) | 21 | 55% | 25 | 69% | 20 | 59% | 66 | 61% |
| Texts (including WhatsApp messages, etc) | 13 | 34% | 15 | 42% | 12 | 35% | 40 | 37% |
| Facebook/other web-based social media platforms | 3 | 8% | 2 | 6% | 2 | 6% | 7 | 6% |

Data are number (%) of pharmacies providing or receiving requests (main row) and number of clients in past week (subsidiary row).
*Indicates significant difference between counties (p<0.05), based on χ2 test or analysis of variance for means.
†A significant proportion of these sales may not have been related to COVD-19 prevention/ treatment.
‡Traditional/alternative/natural remedies included echinacea, chapa shoka, kaluma inhaler, menthol plus balm and home remedies (lemon, papaya steaming).
§Antibiotics included azithromycin (n=55), cephalosporin (n=7), amoxicillin-containing (n=6), other macrolide (n=2); 3 pharmacies did not specify.
¶Other medicines included zinc+ vitamin D (n=4), vitamin B complex (n=1), salbutamol (n=1) and scotts emulsion (n=1); 2 pharmacies did not specify.

colleagues (2%). For specific questions about COVID-19, providers consulted the pharmacy-in-charge (48%), nearby doctor (37%), MoH officials (32%), the internet (5%) and medical representatives (2%).

Interviewees felt that the official announcement of the first case played a major role in awareness creation, though some reported having learnt about the outbreak prior to that, through radio, television and social media. However, providers decried the rapidly changing and sometimes contradictory information, for example *'on whether to use or not to use masks'*.

Fifty-two (48%) pharmacies had at least one staff member trained on COVID-19, with a median (range) of 1[1–8] providers trained. This did not vary by county or by membership in professional associations. Modalities of training included: on-site in the pharmacy (67% of respondents), virtual (54%) and off-site (13%). Virtual trainings were more prevalent in Kisumu compared with Nairobi (76% vs 35%, p=0.04). Training sessions were facilitated by MoH (73%), KPA (51%), PSK (19%), African Medical & Research Foundation (AMREF) (9%) and Kenyatta National Hospital (5%). While some interview participants felt they had received 'a lot' of training, others felt the trainings were not very adequate, particularly with regard to inclusion of updated information in follow-up sessions and the fact that most trainings included only the pharmacy in-charge and not the entire staff.

### External support and linkages
Interviewees expressed appreciation for guidance from the ministry of health, including the pharmacy and poisons board, who '*did some inspection and brought some banners [information materials] on how I can protect myself and the general public.*' The government-led community awareness-creation activities were also hailed as an important complement to pharmacy efforts in educating the public.

However, examples of direct linkages and material support were rare. One interviewee in Nairobi reported that community health workers from a nearby health centre regularly visited the pharmacy to collect information on number of suspect cases seen. Another interviewee in Kisumu described an instance where an ambulance was sent promptly after he reported a patient in critical condition.

More commonly, a feeling of isolation and neglect prevailed, with pharmacies reporting that they were not receiving much support from the government or other external parties with regard to COVID-19, unlike what happens with other public health programmes:

**Table 3**  Handling of simulated clients seeking COVID-19-related services at private retail pharmacies in Kenya

| | Nairobi (N=38) | | Mombasa (N=36) | | Kisumu (N=34) | | Total (N=103) | |
|---|---|---|---|---|---|---|---|---|
| Call outcome | n | % | n | % | n | % | n | % |
| **Provider response to 'Do you have medicines that I can use to protect myself from getting COVID-19?'** | | | | | | | | |
| Asked the client why he/she was asking for the medicines | 14 | 38* | 4 | 14 | 2 | 6 | 20 | 20 |
| Recommended conventional therapies | 11 | 29 | 6 | 19 | 11 | 32 | 28 | 27 |
| Antibiotics | 11 | 29 | 6 | 19 | 10 | 29 | 27 | 26 |
| Azithromycin | 9 | 24 | 6 | 19 | 9 | 26 | 24 | 23 |
| Other antibiotics† | 2 | 5 | 1 | 3 | 4 | 12 | 7 | 7 |
| Painkillers | 1 | 3 | 0 | 0 | 3 | 9 | 4 | 4 |
| Antihistamines | 0 | 0 | 1 | 3 | 2 | 6 | 3 | 3 |
| Recommended alternative therapies | 29 | 76 | 22 | 71 | 22 | 65 | 73 | 71 |
| Vitamin C | 28 | 74 | 20 | 65 | 19 | 56 | 67 | 65 |
| Zinc | 25 | 66* | 10 | 32 | 7 | 21 | 42 | 41 |
| Home remedies‡ | 6 | 16 | 4 | 13 | 6 | 18 | 16 | 16 |
| Vitamin B/D | 4 | 11 | 4 | 13 | 2 | 6 | 10 | 10 |
| Multivitamins | 2 | 5 | 0 | 0 | 3 | 9 | 5 | 5 |
| Other remedies§ | 1 | 3 | 0 | 0 | 1 | 3 | 2 | 2 |
| Gave options on how to get the recommended medicines | 37 | 97 | 28 | 90 | 34 | 100 | 99 | 96 |
| Pick from the pharmacy | 21 | 55 | 7 | 23* | 18 | 53 | 46 | 45 |
| Home delivery | 19 | 50* | 9 | 29 | 4 | 12 | 32 | 31 |
| Send someone to pick | 5 | 13 | 8 | 26 | 4 | 12 | 17 | 17 |
| **Provider response to 'What symptoms should I look out for if I suspect COVID-19 infection?'** | | | | | | | | |
| Mentioned any symptoms | 34 | 89 | 29 | 94 | 30 | 88 | 93 | 90 |
| Fever | 23 | 61 | 20 | 65 | 21 | 62 | 64 | 62 |
| Cough | 13 | 34 | 15 | 48 | 16 | 47 | 44 | 43 |
| Influenza-like symptoms | 15 | 39 | 15 | 48 | 12 | 35 | 42 | 41 |
| Breathing difficulty | 13 | 34 | 12 | 39 | 17 | 50 | 42 | 41 |
| Loss/change of smell or taste | 16 | 42 | 13 | 42 | 11 | 32 | 40 | 39 |
| Sore throat | 15 | 39* | 8 | 26 | 4 | 12 | 27 | 26 |
| Chest pain or congestion | 7 | 18 | 4 | 13 | 12 | 35* | 23 | 22 |
| Headache | 6 | 16 | 4 | 13 | 8 | 24 | 18 | 17 |
| Fatigue, tiredness, malaise | 2 | 5 | 8 | 26* | 4 | 12 | 14 | 14 |
| Reduced appetite | 2 | 5 | 2 | 6 | 7 | 21 | 11 | 11 |
| Body aches/ pains | 0 | 0 | 3 | 10 | 2 | 6 | 5 | 5 |
| Gut symptoms, for example, stomach pain, diarrhoea, etc | 2 | 5 | 0 | 0 | 2 | 6 | 4 | 4 |
| Dizziness | 0 | 0 | 2 | 6 | 0 | 0 | 2 | 2 |
| Weight loss | 0 | 0 | 0 | 0 | 1 | 3 | 1 | 1 |
| **Provider response to 'What should I do if I suspect I have COVID-19 infection?'** | | | | | | | | |
| Immediate actions recommended | | | | | | | | |
| Get tested for COVID-19 | 25 | 66 | 18 | 58 | 16 | 47 | 59 | 57 |
| Visit nearest health facility/go see a doctor | 18 | 47 | 13 | 42 | 24 | 71* | 55 | 53 |
| Self-quarantine/isolate at home | 3 | 8 | 9 | 29* | 4 | 12 | 16 | 16 |
| Take medications¶ | 3 | 8 | 2 | 6 | 3 | 9 | 8 | 8 |
| Check body temperature | 2 | 5 | 1 | 3 | 2 | 6 | 5 | 5 |
| Check oxygen levels | 0 | 0 | 1 | 3 | 1 | 3 | 2 | 2 |
| Check information online | 0 | 0 | 1 | 3 | 1 | 3 | 2 | 2 |
| Call 719** | 2 | 5 | 0 | 0 | 0 | 0 | 2 | 2 |

**Table 3** Continued

| Call outcome | Nairobi (N=38) | | Mombasa (N=36) | | Kisumu (N=34) | | Total (N=103) | |
|---|---|---|---|---|---|---|---|---|
| | n | % | n | % | n | % | n | % |
| **Supplementary actions recommended** | | | | | | | | |
| Wear a mask | 23 | 61 | 16 | 52 | 20 | 59 | 59 | 57 |
| Maintain physical distance | 21 | 55 | 15 | 48 | 18 | 53 | 54 | 52 |
| Observe hand hygiene | 7 | 18 | 5 | 16 | 7 | 21 | 19 | 18 |
| Eat fruits | 5 | 13 | 5 | 16 | 5 | 15 | 15 | 15 |
| Eat well†† | 2 | 5 | 3 | 10 | 1 | 3 | 6 | 6 |
| Take lots of fluids | 1 | 3 | 1 | 3 | 3 | 9 | 5 | 5 |
| Live well‡‡ | 0 | 0 | 0 | 0 | 4 | 12 | 4 | 4 |
| Other advice§§ | 3 | 8 | 1 | 3 | 2 | 6 | 6 | 6 |
| **Caller assessment of the interaction** | | | | | | | | |
| How friendly (approachable) was the person who served you? | | | | | | | | |
| Very unfriendly | 1 | 3 | 0 | 0 | 0 | 0 | 1 | 1 |
| Unfriendly | 4 | 11 | 3 | 10 | 4 | 12 | 11 | 11 |
| Friendly | 20 | 53 | 22 | 71 | 22 | 65 | 64 | 62 |
| Very friendly | 13 | 34 | 6 | 19 | 8 | 24 | 27 | 26 |
| How enthusiastic was the provider about providing the service? | | | | | | | | |
| Very unenthusiastic | 1 | 3 | 0 | 0 | 0 | 0 | 1 | 1 |
| Unenthusiastic | 4 | 11 | 4 | 13 | 7 | 21 | 15 | 15 |
| Enthusiastic | 22 | 58 | 22 | 71 | 21 | 62 | 65 | 63 |
| Very enthusiastic | 11 | 29 | 5 | 16 | 6 | 18 | 22 | 21 |
| Did the provider say or do anything that you found particularly helpful or nice¶¶? | 19 | 50 | 9 | 29 | 14 | 41 | 42 | 41 |
| Did the provider say or do anything that you found particularly NOT helpful or NOT nice***? | 4 | 11 | 2 | 6 | 7 | 21 | 13 | 13 |

*Indicates significant difference between counties (p<0.05), based on χ2 test.
†Other antibiotics included Augmentin, Amoxil, ceftriaxone, etc.
‡Home remedies included ginger/lemon/hot water drink, hot soup/ water, garlic, steam, frequent hot baths.
§Other alternative therapies included ginsomin and omega-3.
¶Medications for treatment included: azithromycin (3%), other antibiotics (4%), painkillers (2%), steroids (1%), home remedies (1%).
**719 is the government hotline for COVID-19 reporting and information-seeking.
††Eat well: eat healthy, eat vegetables, etc.
‡‡Live well: exercise, enough rest, fresh air, don't panic, avoid stress, etc.
§§Other advice: don't touch face, avoid handshakes, do not take any medicines, don't reuse mask, etc.
¶¶Things that were deemed particularly helpful or nice related mainly to: reassurance that infection risk is low if prevention measures are followed; high chance of recovery in case of infection; detailed information on what to do or where to seek further help; and follow-up calls in cases where the provider was busy at the time of the initial call.
***Things that were deemed not helpful or not nice related mainly to: calls being hang up; inattentiveness or being in a hurry to finish the call; and dismissiveness (eg, 'don't you watch news? The symptoms are public knowledge!' and 'we don't have those medicines you are asking for, jaribu kwingine [try elsewhere])'.

So far being a private facility, we are not receiving any financial support from the organizations in COVID related issues. We don't receive that apart from PS Kenya [NGO supporting TB and HIV programs].

- Male pharmaceutical technologist, Kisumu, 301

### Confidence, motivation and coping mechanisms

Majority (87%) of questionnaire respondents reported feeling confident (68%) or very confident (19%) about providing COVID-19-related services. Over half felt prepared (51%) or very prepared (7%) about handling a client who presents to the pharmacy with COVID-19 symptoms.

Motivation for supporting delivery of COVID-19 services included: 'to help people protect themselves or prevent transmission (75%), 'It's part of my job/ I am required to do it' (14%), encouragement and support from professional associations (7%), 'to boost the profile of the pharmacy' (2%), and 'to make money' (1%). When asked how much they felt their personal role in the COVID-19 response was valued by the community, the majority stated highly valued (68%) or very highly valued (11%).

In addition to the operational adjustments described above, interviewees described other coping mechanisms such putting up a shade outside the pharmacy to prevent crowding inside and improvisation of hand sanitisers when not available from suppliers. Some interviewees also reported going the extra mile sometimes, for example, by providing free masks to clients.

**Table 4** COVID-19 infection prevention during service provision by private retail pharmacies in Kenya

| Infection prevention measures | Nairobi (N=38) | | Mombasa (N=36) | | Kisumu (N=34) | | Total (N=108) | |
|---|---|---|---|---|---|---|---|---|
| | n | % | n | % | n | % | n | % |
| **Among staff** | | | | | | | | |
| Physical distancing* | 25 | 66 | 26 | 72 | 27 | 79 | 78 | 72 |
| Temperature monitoring | 13 | 34 | 14 | 39 | 13 | 38 | 40 | 37 |
| Hand hygiene | 35 | 92 | 33 | 92 | 32 | 94 | 100 | 93 |
| Disinfection of surfaces | 29 | 76 | 30 | 83 | 27 | 79 | 86 | 80 |
| Face masks† | 34 | 89 | 30 | 83 | 30 | 88 | 94 | 87 |
| Goggles | 0 | 0 | 1 | 3 | 1 | 3 | 2 | 2 |
| Ingestion of immune boosters | 1 | 3 | 0 | 0 | 0 | 0 | 1 | 1 |
| **Among clients** | | | | | | | | |
| Physical distancing* | 33 | 87% | 31 | 86% | 30 | 88% | 94 | 87% |
| Temperature monitoring | 13 | 34% | 9 | 25% | 11 | 32% | 33 | 31% |
| Hand hygiene‡ | 36 | 95% | 33 | 92% | 30 | 88% | 99 | 92% |
| Face masks | 27 | 71% | 32 | 89% | 27 | 79% | 86 | 80% |
| Non-touch payment methods§ | 14 | 37% | 16 | 44% | 10 | 29% | 40 | 37% |
| Information materials and messages¶ | 12 | 32% | 13 | 36% | 11 | 32% | 36 | 33% |

*Physical distancing was achieved through various means including: reducing number of staff and working in shifts, ensuring a distance of 1.5–3 m between staff and clients through the use of posters and floor/seat markings, serving one client at a time or other crowd control measures, ribbons/ropes/glass/ metal-grill/ wooden barriers between client and provider, and home delivery of products. One pharmacy noted that 'patients with cough and related symptoms were always first referred to the health centre for COVID-19 testing before being served further'.
†Majority of pharmacies (95%) were using surgical/medical masks; only 6% reported use of cloth masks.
‡Hand hygiene measures among clients included: handwashing, hand sanitiser and 'no hand-shakes' policy.
§Non-touch payment methods included: mobile money and credit/ debit cards; one pharmacy reported disinfection of cash.
¶Messages were delivered through: posters (21%), leaflets (5%), television screens (2%) and audio announcements (1%).

### Provider suggestions and recommendations

Seventy-six (70%) questionnaire respondents highlighted contributions that they felt pharmacies can uniquely make to the national response, including: provision of information and counselling to clients (64%), distribution of preventive materials (21%), triage and referral of suspect cases (17%), surveillance of infection rates and community adherence to preventive measures (14%), provision of preventive and curative therapies (12%) and maintenance of long-term therapy (3%). Pharmacies' location within communities was cited as a strength when it comes to surveillance of client behaviour and community awareness creation.

To facilitate these inputs, pharmacies suggested additional support that they would require from the government and other external parties, including: training (34% of respondents), free or subsidised preventive materials such as masks, sanitisers and thermo guns for own use and distribution to clients (34%), supply management such as price and quality controls (15%), clear guidelines and referral systems (9%), community awareness creation (7%), information materials (6%), financial support such as loan repayment holidays and medical insurance (6%) and testing services (5%).

These suggestions were echoed in interviews. Postmarketing surveillance to ensure quality of products in the market and a streamlined process to capture and transmit information back to the government were pinpointed as

key gaps in the pandemic response. Overall, there were earnest calls for more deliberate engagement, oversight and support supervision by the government and other implementing partners:

> If the government could put us, the pharmacists and pharm techs in the you know, treat us like people who are fighting this virus and facilitate this thing it would be a major shift, it would be a major shift.
>
> - Male pharmacy in charge, Mombasa, 205

### DISCUSSION

In this study among private retail pharmacies (community pharmacies) in Kenya, we describe the impact of the COVID-19 pandemic on operations, COVID-19-related products and services being provided, infection prevention during service provision, and provider suggestions and recommendations on how to improve pandemic preparedness and response.

The initial weeks after the pandemic reached Kenya were characterised by fear and panic among providers and a surge in client flow. Broadly speaking, the experiences documented in our study can be captured with the apt words of Austin and Gregory in their paper describing the experiences of community pharmacists in Ontario, Canada, viz: 'the scale and magnitude of the COVID-19 pandemic defied imagination or preplanning, yet

healthcare professionals like pharmacists were required to continue to provide services and care to patients—and in many cases, expand their repertoire of clinical skills to assume new and even more challenging responsibilities'.[15]

The initial surge in demand waned rapidly, with client flow dipping below prepandemic levels within a few weeks and remaining depressed until the time of data collection, 7 months later. This was surprising as we had hypothesised, before the study, that pharmacies would experience an overall increase in demand as patients redirected their care-seeking from health facilities for fear of exposure to infection and quarantine. Participants attributed the decrease in demand mainly to the movement restrictions instituted by government. However, they also suggested—ironically—that clients, especially those with influenza-like symptoms, may have avoided pharmacies for fear of being referred to public facilities and subsequently quarantined.

Our findings clearly show that community pharmacies in Kenya were undertaking potentially impactful activities to support the response to the COVID-19 pandemic. Despite a feeling of disconnection from the government-led national response, providers expressed great readiness and motivation for provision of COVID-19-related services, similar to providers in Qatar,[30] the USA[31] and other settings around the world.[32] The services pharmacies in our study were providing are generally similar to those suggested by experts in the field,[22 33–35] as well as those being provided in other low-to-middle-income countries[36–39] and in high-income settings.[30 40 41]

However, management of drug shortages and prevention of stockpiling (hoarding) by clients[22 30 33] were reported only as challenges and not as roles that Kenyan pharmacies were playing or could play. Additionally, while 60% of questionnaire respondents said they were receiving queries and providing information and counselling over the phone, other remote services such as online prescribing and home delivery of medications[22 36] were not being widely used or mentioned as potential additional roles. Participants in our study, being front-line workers, may have deemed the carrying out of these innovations as being beyond the sphere of their influence. Similarly, COVID-19 vaccine administration was not mentioned as a potential role, understandably perhaps because a vaccine was not available at the time. But pharmacies in Kenya are actively involved in provision of other vaccines,[42 43] hence it is reasonable to hypothesise that COVID-19 vaccines could be delivered through private pharmacies in the future. Interviews with policymakers and captains of industry may be required to probe these topics further.

The finding that no pharmacy was providing COVID-19 testing was unsurprising since no approved POC tests were available at the time of the study. The more interesting finding was that significant demand and support for pharmacy-based testing existed, similar to what studies in the Middle East[44] and the USA[31] have found. In this context and given the fact that at least three rapid POC tests have since been approved in Kenya,[45] there is an urgent need to start developing policies to guide implementation. This will obviate the emergence of an unco-ordinated and potentially dangerous service, like what we saw in 2016 with HIV testing before policy guidelines were issued the following year.[46 47]

For treatment of COVID-19 symptoms, most pharmacies recommended alternative therapies and nutritional supplements such as vitamin C and zinc, and only about one-third suggested conventional therapies such as antibiotics which have no proven efficacy.[13 14] This finding is similar to that of a small study in Nairobi in May 2020 which found that despite increased requests for antimalarials and antibiotics, pharmacists recommended alternative therapies.[48] A similar finding was also obtained in a multinational survey of Asian pharmacists.[35] While alternative therapies and nutritional supplements are themselves not evidence-based (as no rigorous trials have proven their efficacy against SARS-CoV-2), they have a better safety profile and their useful effects are more biologically plausible since they have been used for ages against the influenza virus, itself a coronavirus. These findings are therefore encouraging. However, among Egyptian pharmacies, the antibiotic azithromycin was given to about 40% of presumptive COVID-19 patients with mild to moderate symptoms, although based on physician recommendation.[49] Further research in different jurisdictions may be needed to describe and contrast the use of antibiotics and other unproven COVID-19 therapies, in order to inform the promotion of rational drug use and the prevention of antimicrobial resistance.

Our study identified other important gaps in community pharmacies' preparedness and response to the pandemic. Notably, pharmacies did not seem to have any well-defined operational linkages to other units of the healthcare system, including, for example, the emergency operations centre at the ministry of health. Besides, while only about half of pharmacies had at least one staff member trained on COVID-19, some of those trained felt they had received a lot of training, suggesting a 'network effect' where some providers were receiving many trainings while others received none.

Specific programmes will be required to address these gaps and enhance pandemic preparedness. Templates for public-private cooperation in pandemic responses do exist from other settings. Ung, for example, reports how in one administrative region on the south-east coast of China, pre-existing public–private partnerships between the local government and community pharmacies ensured rapid provision and prevented hoarding of face masks through a government-led 'Guaranteed Mask Supply for Macao Residents Scheme'.[50] A qualitative study in Canada, identified key features that predicted community pharmacists' resilience during the early period of the pandemic, including: use of and comfort with technology; early adoption of corporate and professional guidance; emphasis on task-focus rather than multitasking; and provision of personal protective equipment.[15] Roll-out

of pharmacy-based COVID-19 programmes should be accompanied with implementation studies to better uncover aspects of capacity, preparedness and resilience.

Our study had a number of strengths. First, we took advantage of information collected in a recent study to speed up recruitment and data collection. Second, we used multiple quantitative and qualitative methods to enhance the comprehensiveness and reliability of the findings. Of note, we used SCs to objectively assess provider practices, an approach that has been shown to be superior to methods that rely on real patients, such as exit interviews, chart abstraction and direct observation[51]; and is gaining acceptance as a gold standard for the measurement of clinical practice quality around the world.[52] Our study also had some weaknesses. We observed a significant variation in detection rate by individual fieldworkers, suggesting variation in the way different fieldworkers enacted the simulation. However, detection rate did not vary across counties (perhaps because fieldworkers were scheduled across counties) and outcomes did not vary by fieldworker, hence this bias is unlikely to have had an effect on our comparisons across counties. Another weakness of our study was the somewhat low response rate (55%), considering that we targeted pharmacies that had participated in a similar study less than a year previously. Compared with non-participating pharmacies, participating pharmacies had features suggestive of higher operational capacity and interest in public health interventions. While this limits generalisability of our findings, it can also be looked at in positive light. We propose that pharmacies with these characteristics could be identified for inclusion in a 'practice and learning network' that could provide a platform for testing implementation strategies for various public health interventions.[53–55] Scale up of such interventions would then target similar pharmacies and include strategies for enhancing operational capacity and interest in public health among the other pharmacies.

In conclusion, we found that private retail pharmacies in Kenya were actively contributing to the COVID-19 response despite an apparent disconnection from the national response programme. More deliberate engagement, support and linkages are required. Notably, there is an urgent need to develop policy guidelines for pharmacy-based COVID-19 testing, a service that is clearly needed and which could greatly increase test coverage. Roll-out of this and other pharmacy-based COVID-19 programmes should be accompanied with implementation research in order to inform current and future pandemic responses.

**Acknowledgements** We acknowledge the contributions of all participating pharmacy service providers. We thank James Wafula for supporting database development and data management; and Langat Justus, Lucy Omumbo, Milkah Auka and Rodgers Ayoma for supporting the simulated client survey.

**Collaborators** PSK COVID19 Response Taskforce: Collins Jaguga, Davy Odhiambo, Elizabeth Ominde-Ogaja, Eric Seda, Mercy Maina, Michael Mungoma, Nadia Butt, Sultani Matendechero, Sylvia Opanga, Titus Kahiga.

**Contributors** PMM is the guarantor: accepts full responsibility for the work and the conduct of the study, had access to the data, and controlled the decision to publish.

Conceived the study: PMM and EB. Developed protocol: PMM, DM, SM and EB. Performed the experiments: PMM and AM; Analysed the data: PMM, AM, JN. Wrote the manuscript: PMM, AM, DM, JN, SM and EB. Approved the final manuscript: PMM, AM, DM, JN, SM and EB.

**Funding** This work was supported by Wellcome Trust (#211353/Z/18/Z). For the purpose of open access, the author has applied a CC-BY public copyright licence to any author accepted manuscript version arising from this submission. The KWTRP at the Centre for Geographical Medicine Research-Nairobi is supported by core funding from the Wellcome Trust (#077092). This report was published with the permission of Director KEMRI CGMRC.

**Competing interests** None declared.

**Patient and public involvement** Patients and/or the public were involved in the design, or conduct, or reporting, or dissemination plans of this research. Refer to the Methods section for further details.

**Patient consent for publication** Consent obtained directly from patient(s)

**Ethics approval** This study involves human participants and was approved by Kenya Medical Research Institute (KEMRI) Scientific and Ethics Review Unit (KEMRI/SERU/CGMR-C/208/4080). Participants gave informed consent to participate in the study before taking part.

**Provenance and peer review** Not commissioned; externally peer reviewed.

**Data availability statement** Data are available in a public, open access repository. Underlying data can be accessed in the Harvard Dataverse using the following link:https://doi.org/10.7910/DVN/XTLUFP.

**ORCID iDs**
Peter Mwangi Mugo http://orcid.org/0000-0002-1808-3292
Jacinta Nzinga http://orcid.org/0000-0001-8394-8857

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
