## [Reviewer comments · BMJ Open]

ARTICLE DETAILS

TITLE (PROVISIONAL)	Experiences of and response to the coronavirus disease 2019 (COVID-19) pandemic at private retail pharmacies in Kenya: a mixed methods study
AUTHORS	Mugo, Peter; Mumbi, Audrey; Munene, Daniella; Nzinga, Jacinta; Molyneux, Sassy; Barasa, Edwine; Taskforce, COVID19

VERSION 1 – REVIEW

REVIEWER	Alshammari, Thamir King Saud University, Medication Safety Research Chair
REVIEW RETURNED	15-Jan-2022

GENERAL COMMENTS	Thank you for giving me the chance to review this manuscript. The manuscript entitled “Experiences and response to the coronavirus disease 2019 (COVID-19) pandemic at private retail pharmacies in Kenya: a mixed-methods study .” The study idea is great and important, it also has good results, however, there are some limitations that need to be addressed. Here are my comments. General comment The study was conducted well, however, the response rate was one of the main drawbacks in this study and since there was no power calculation, we are not sure if this response rate is enough. Also, I believe there is a need to discuss the methods in more detail. Introduction Page 4, reviewer’s comment, please elaborate on the community pharmacies in Kenya in detail and the services they provided Page 4, line 51, reviewer’s comment, please make the aim of your study is what actually you were aiming to, i.e., the aim of your study was to assess “.....preparedness and response to the COVID-19 pandemic at private retail pharmacies in Kenya to.....” Methods Page 5, line 15, reviewer’s comment, why these specific counties? Why not more to have more coverage of covid19 cases? Page 5, line 15, reviewer’s comment, what is the process of contacting the pharmacies? How many times if there is no response? Please elaborate more on this
---

	Page 5, line 21, reviewer's comment, why targeted HIV participated pharmacies? Why not in general to avoid selection bias? Page 5, line 22, reviewer's comment, what do you mean by a manuscript in preparation? Page 5, line 25, reviewer's comment, degree holders mean bachelor degree? If so please be specific Page 6, line 12, reviewer's comment, how did you end up with your sample size? Did you do any power calculations? Please provide your power calculations. Page 6, line 17, reviewer's comment, please provide details on your regression analysis and how did you do it Results Reviewer's comment, the results look good and important information is available please arrange them by bullets for the readers to follow with you Page 7, line 1, reviewer's comment, please add the Cis for the aOR Page 7, line 15, Reviewer's comment, why only 18 were interviewed? Did you do any process to increase the number of interviewed pharmacies? Discussion Reviewer's comment, the discussion is good, yet, some results were not discussed and justify them in the section, please make sure to discuss all your results.
--	---

REVIEWER	Hoti, Kreshnik Curtin University, School of Pharmacy
REVIEW RETURNED	24-Jan-2022

GENERAL COMMENTS	In general, this is well written manuscript providing important insight into experiences and response to COVID-19 pandemic in Kenyan community pharmacies. Combination of various data collection techniques strengthens the paper. To reach its full publishing potential, I would suggest the following to authors:  • Please double check English grammar throughout the manuscript • In objectives (abstract) I would suggest authors reword the research question in a way that it is not provided as a question but as an objective; • This sentence is not clear "Fifty-nine pharmacies (55% [95% CI 45-65%]) reported ever receiving a client.."; Suggest rewording. • Results provided in the abstract, do not seem to be sensitive to clients and pharmacies considering that data was collected from clients as well.
---

	 • Under participants (abstract), authors have written ‘survey respondents’ . Were they clients as well or pharmacists? Please clarify. • In the introduction section, last paragraph, please provide more argumentation regarding the need for this study; • In the methods section, please provide more details regarding how questionnaires were distributed e.g. was there a follow up? Where these all online questionnaires... • Authors state randomization of pharmacies, please provide more details in terms of how this was done • Noted interviews were conducted by study authors. Where there any bias mitigation strategies employed during the process. If so, please provide, if not please list under limitations section. • Was there any theoretical framework used for the analysis of qualitative data? • Noted multivariate regression stated. These results have not been provided in the abstract. Would suggest authors provide key results from these analysis in the abstract section as well. • I would suggest authors review the limitations section in the discussion. Currently only one weakness has been provided, regarding the response rate. (55%), which otherwise is not a low response rate anyway. I believe there are additional limitations that merit recognition by authors.
--	--

VERSION 1 – AUTHOR RESPONSE

REVIEWER #1:	
General comment	
The study was conducted well, however, the response rate was one of the main drawbacks in this study and since there was no power calculation, we are not sure if this response rate is enough. Also, I believe there is a need to discuss the methods in more detail.	We thank the reviewer for recognizing the rigorous conduct of the study. We have addressed the comment on power under the specific comment below and fleshed out the methods section, also in response to other reviewer and editorial comments.
Introduction	
Page 4, please elaborate on the community pharmacies in Kenya in detail and the services they provided	We have added a description of the community pharmacy sector in Kenya under “Methods> Study settings and populations”
Page 4, line 51, please make the aim of your study is what actually you were aiming to, i.e., the aim of your study was to assess “.....preparedness and response to the COVID-19 pandemic at private retail pharmacies in Kenya to.....”	We have rephrased the aim of the study as recommended.
Methods	
Page 5, line 15, why these specific counties? Why not more to have more coverage of covid19 cases?	In the section “Methods> Study settings and populations”, it was clarified that counties were selected purposively to represent the main urban centres. We have added that these counties have a “high concentration of pharmacies and COVID-19 cases.”
Page 5, line 15, what is the process of contacting the pharmacies? How many times if there is no response? Please elaborate more on this	In the section “Data collection”, we have added that “we used previously obtained phone numbers to

	contact sampled pharmacies, obtain informed consent and share a link to an online questionnaire.”
Page 5, line 21, why targeted HIV participated pharmacies? Why not in general to avoid selection bias?	We chose to do the current study in the same pharmacies as the previous study in order to speed up data collection, by minimizing the time spent on recruitment. However, we don't believe this created bias since as clarified under “Methods> Study settings and populations”, “Pharmacies had been selected randomly from the list of licensed pharmacies.”
Page 5, line 22, what do you mean by a manuscript in preparation?	We have deleted this potentially confusing statement.
Page 5, line 25, degree holders mean bachelor degree? If so please be specific	We have clarified that this means “bachelor's” degree
Page 6, line 12, how did you end up with your sample size? Did you do any power calculations? Please provide your power calculations.	Under “Methods> Study settings and populations”, we have added that “The sample size for the PhP study (n=195) had been calculated to provide 80% power to detect associations at the 5% significance level, assuming a binary dependent variable split at 70% and 50% across a binary predictor variable. The sample size achieved in the current study (n=108) gives 57% power for similar parameters and assumptions, and 80% power for a 75%-50% split.”
Page 6, line 17, please provide details on your regression analysis and how did you do it	We have added that “Variables with $p < 0.10$ in bivariate analysis were included in the multivariable model and those with $p < 0.05$ in the multivariable model were considered to be statistically significant.”
Results	
Reviewer's comment, the results look good and important information is available please arrange them by bullets for the readers to follow with you	We do not fully understand what is meant by “arrange by bullets”. However, since the journal requires the main sections of the paper be presented in prose, we have maintained this style. The results section is arranged by thematic sub-sections which we believe the reader will find easy to follow.
Page 7, line 1, please add the CIs for the aOR	Given that p-values are provided and recognizing that these are supplementary analyses, we do not believe there is value in adding CIs.
Page 7, line 15, why only 18 were interviewed? Did you do any process to increase the number of interviewed pharmacies?	A sample size of 18 was selected based on guidelines (Silverman 2013, Sim et al 2018) and previous experience with similar research questions. As is typical of qualitative designs, sample sizes are provisional and will depend on reaching a point of saturation, where no new key themes emerge. In supplementary file E (COREQ checklist) item 22, it was clarified that “to expedite data collection, all 18 interviews were completed before starting analysis or checking for saturation.” We have added (in the checklist) that “Saturation was reached after 14 interviews, but all 18 interviews were included in analysis.”
Discussion	

The discussion is good, yet, some results were not discussed and justify them in the section, please make sure to discuss all your results.	Due to word limit, we discussed only the main and most interesting findings.
REVIEWER #2:	
General comments	
In general, this is well written manuscript providing important insight into experiences and response to COVID-19 pandemic in Kenyan community pharmacies. Combination of various data collection techniques strengthens the paper.	We thank the reviewer for recognizing the strengths and importance of the study.
Please double check English grammar throughout the manuscript	We have reviewed the grammar and made corrections as appropriate (not tracked)
Abstract	
In objectives (abstract) I would suggest authors reword the research question in a way that it is not provided as a question but as an objective;	We have rephrased the objective as recommended.
This sentence is not clear “Fifty-nine pharmacies (55% [95% CI 45-65%]) reported ever receiving a client..”; Suggest rewording.	We have reworded the sentence to clarify.
Results provided in the abstract, do not seem to be sensitive to clients and pharmacies considering that data was collected from clients as well.	We have edited the sentence that starts “For treatment...” to clarify that this finding is based on the simulated client survey.
Under participants (abstract), authors have written ‘survey respondents’ . Were they clients as well or pharmacists? Please clarify.	We have clarified that the study was among service providers only. Simulated “client” survey is an evaluation method and does not mean that real clients were surveyed. This is made clear in the data collection section of the main text.
Introduction	
In the introduction section, last paragraph, please provide more argumentation regarding the need for this study;	We have clarified that the study had “an overall goal of identifying strategies for maximising the contribution of pharmacies to the national response.”
Methods	
In the methods section, please provide more details regarding how questionnaires were distributed e.g. was there a follow up? Where these all online questionnaires...	We have clarified that “We used previously obtained phone numbers to contact sampled pharmacies, obtain informed consent and share a link to an online questionnaire. One structured online questionnaire, taking about 20-30 minutes, was completed by each participating pharmacy (Supplementary file A). Up to three reminder phone calls were made.”
Authors state randomization of pharmacies, please provide more details in terms of how this was done	There was no statement on “randomization of pharmacies” as this was not a trial. Rather, it is stated that pharmacies were randomly selected.
Noted interviews were conducted by study authors. Where there any bias mitigation strategies employed during the process. If so, please provide, if not please list under limitations section.	Due to word limit, this is reported in supplementary file E (COREQ checklist), under “Domain 1: Research team and reflexivity”
Was there any theoretical framework used for the analysis of qualitative data?	Under “Data management and analysis”, it is clarified that “Analysis followed a thematic approach with initial categories based on the interview guide and emergent themes integrated in a second round of coding.” Further details are provided in

	supplementary file E (COREQ checklist), under “Domain 3: analysis and findings”
Results	
Noted multivariate regression stated. These results have not been provided in the abstract. Would suggest authors provide key results from these analysis in the abstract section as well.	We have clarified that the modelling mentioned was exploratory in nature and was not addressing the key research question. Besides, there were no findings from the modelling that were interesting enough to include in the abstract.
Discussion	
I would suggest authors review the limitations section in the discussion. Currently only one weakness has been provided, regarding the response rate. (55%), which otherwise is not a low response rate anyway. I believe there are additional limitations that merit recognition by authors.	We could not think of any additional limitations.

VERSION 2 – REVIEW

REVIEWER	Alshammari, Thamir King Saud University, Medication Safety Research Chair
REVIEW RETURNED	21-Mar-2022

GENERAL COMMENTS	Thank you to the authors since they reply to most of the comments, however, there are some comments were not addressed well and not convincing reply as a scientific reply, for instance, How did you end up with your sample size? Did you do any power calculations? Please provide your power calculations. Please add the Cis for the aOR
--

REVIEWER	Hoti, Kreshnik Curtin University, School of Pharmacy
REVIEW RETURNED	12-Apr-2022

GENERAL COMMENTS	The paper has improved and many of the suggestions have been sufficiently addressed. However, there are a couple of remaining issues which I believe would benefit this paper before it can be accepted for publication. In they responses, authors state that the questionnaire 'was completed by each participating pharmacy'. Please clarify weather pharmacists in pharmacies completed the questionnaire or was this any staff member. This is important for readers. In query related to randomization, authors have responded : 'There was no statement on “randomization of pharmacies” as this was not a trial. Random selected implies that all potential participants have an equal chance of being selected. Therefore, authors need to clarify how was this done, for example did they use an electronic randomizer to ensure random selection. Please review if by 'randomly selected' it was actually 'purposive sampling' - which is very different.
---

	To the suggestion about reviewing the need to recognize more limitations, authors have answered 'We could not think of any additional limitations.' I believe this can be revisited. For example, there are limitations when using simulated clients as opposed to using real clients, seeking COVID-19 preventative therapies. Authors have chosen pharmacies which had previously been involved with HIV prevention study. Would this have potentially affected the responses considering that those pharmacies could be more trained/prepared, compared to the ones who were not chosen. The paper would benefit from recognizing these.
--	--

VERSION 2 – AUTHOR RESPONSE

REVIEWER #1:	
General comment	
Thank you to the authors since they reply to most of the comments, however, there are some comments were not addressed well and not convincing reply as a scientific reply, for instance:	We thank the reviewer for this further feedback
How did you end up with your sample size? Did you do any power calculations? Please provide your power calculations.	In the first revision, we had indicated under “Methods> Study settings and populations [3 rd paragraph]” that “The sample size for the PHP study (n=195) had been calculated to provide 80% power to detect associations at the 5% significance level, assuming a binary dependent variable split at 70% and 50% across a binary predictor variable. The sample size achieved in the current study (n=108) gives 57% power for similar parameters and assumptions, and 80% power for a 75%-50% split.”
Please add the CIs for the aOR	We have added confidence intervals for the adjusted odds ratios
REVIEWER #2:	
The paper has improved and many of the suggestions have been sufficiently addressed. However, there are a couple of remaining issues which I believe would benefit this paper before it can be accepted for publication.	We thank the reviewer for this further feedback
In the responses, authors state that the questionnaire ‘was completed by each participating pharmacy’. Please clarify whether pharmacists in pharmacies completed the questionnaire or was this any staff member. This is important for readers.	In the first revision, we had indicated under “Methods> Study settings and populations [3 rd paragraph]” that “Respondents were pharmacists (bachelor’s degree holders) and pharmaceutical technologists (diploma holders).” We have added that “The preferred questionnaire respondent was the pharmacy-in-charge, the most experienced staff member or the most highly educated, in that order.” In the results section under “Response rate and characteristics of participants”, we indicate that “The majority of questionnaire respondents were

	male, below 40 years, and had a pharmacy diploma, and provide details of questionnaire respondents in Supplementary table Ib.”
In query related to randomization, authors have responded : 'There was no statement on “randomization of pharmacies” as this was not a trial. Random selected implies that all potential participants have an equal chance of being selected. Therefore, authors need to clarify how was this done, for example did they use an electronic randomizer to ensure random selection. Please review if by 'randomly selected' it was actually 'purposive sampling' - which is very different.	We have clarified indicated under “Methods> Study settings and populations [3 rd paragraph]” that “Pharmacies had been selected through stratified random sampling where the required number of pharmacies for each county was picked by an independent data manager from the list of licensed pharmacies for that county using the Excel® Rand() function.”
To the suggestion about reviewing the need to recognize more limitations, authors have answered 'We could not think of any additional limitations.' I believe this can be revisited. For example, there are limitations when using simulated clients as opposed to using real clients, seeking COVID-19 preventative therapies. Authors have chosen pharmacies which had previously been involved with HIV prevention study. Would this have potentially affected the responses considering that those pharmacies could be more trained/prepared, compared to the ones who were not chosen. The paper would benefit from recognizing these.	We thank the reviewer for pointing out these possible limitations. With regard to assessing practice and measuring quality, we note that the simulated client method has been shown to be superior to methods that rely on real patients, such as exit interviews, chart abstraction, and direct observation. We have added a statement to this effect and a reference that discusses these issues in detail (see ref #51). We have also added a possible weakness of our implementation of the simulated client method, viz “We observed a significant variation in detection rate by individual fieldworkers, suggesting variation in the way different fieldworkers enacted the simulation...” With regard to selection bias, we had acknowledged in the previous version the possibility of limited generalizability, viz, “Compared to non-participating pharmacies, participating pharmacies had features suggestive of higher operational capacity and interest in public health interventions. While this limits generalizability of our findings, it can also be looked at in positive light...”

VERSION 3 – REVIEW

REVIEWER	Alshammari, Thamer King Saud University, Medication Safety Research Chair
REVIEW RETURNED	17-May-2022
GENERAL COMMENTS	Thanks for your clarifications. No further comments